# Adherence to iron and folate supplementation and associated factors among women attending antenatal care in public health facilities at Covid-19 pandemic in Ethiopia

**Arayasillase Assegid Tefera[1], Neil Abdurashid Ibrahim[2]\*, Abdurezaq Adem Umer[3]**

**1** HCS-RFSA Health and Nutrition Coordinator Dire-Dawa Branch, Dire-Dawa, Ethiopia, **2** Department of Midwifery, College of Medicine and Health Science, Dire-Dawa University, Dire-Dawa, Ethiopia, **3** Public Health Department, College of Medicine and Health Science, Dire-Dawa University, Dire-Dawa, Ethiopia

\* firdowsayuzarsif@gmail.com

**Data Availability Statement:** All relevant data are within the paper.

## Abstract

Adhesion is the degree to which a patient complies with treatment recommendations made by a health care professional. The majority of pregnant women worldwide don't get the recommended amounts of iron and folic acid (30 to 60 mg of iron and 400 g of folic acid/day for 6 months) pregnant women are more likely to develop iron- and folic acid deficiency anemia. For iron and folate supplementation programs to be effective in Ethiopia, adherence is a significant issue. So, this study aimed to evaluate the level and barriers preventing women receiving antenatal care from taking iron and folate supplements. A phenomenological qualitative study design was added to a facility-based cross-sectional study. The sample was established using a double population proportion formula. For the quantitative and qualitative study, 308 pregnant women and the focal points for the health facilities were chosen at random, using systematic random sampling and purposive sampling techniques. For the quantitative study, a face-to-face interviewer-guided, pre-tested structured questionnaire was used; for the qualitative study, a semi-structured questionnaire was used. Data was entered twice, cross-checked by comparing the two separate entries in Epi Data version 7.2.2.6, and exported to SPSS version 25 for analysis. COR and AOR with 95% CI are used to evaluate the relationship between variables and control for confounding factors. Statistical significance was declared at a p-value < 0.05. All, 308 (100%) participants were involved. 56.5% of pregnant women attending an ANC clinic (95% CI: 51%–62.2%) adhered to Iron and folate supplementation. Mothers with primary education], Urban residents mothers, Mother who had four or more ANC visits, mothers who had registered for their first ANC at early gestational age, mothers who had good awareness about birth defects were independent predictors of adherence to Iron and folate supplementation. In our study, adherence to iron and folate supplementation was low relative to previous research findings. Promoting early and frequent ANC visits and improving pregnant women's awareness of anemia and birth defects through education is necessary to increase the adherence status.

**Funding:** The authors received no specific funding for this work.

**Competing interests:** The authors have declared that no competing interests exist.

**Abbreviations:** ANC, Antenatal care; IFAS, Iron Folic Acid Supplement; Hgb, Haemoglobin; HC, Health centre; RBC, Red Blood Cell; GA, gestational age; SSA, Sub-Saharan Africa; WHO, World Health Organization.

# Background

WHO defined anemia as "a condition in which the quantity or oxygen-carrying capacity of red blood cells is insufficient to meet the physiological needs" [1]. A mineral called iron is a crucial part of the RBC protein Hgb, which transports oxygen from the lungs to the tissues [2]. Especially during pregnancy and infancy, our bodies require folic acid for the synthesis, repair, methylation, and cell division of DNA as well as for the production of healthy RBCs and the prevention of anemia [3]. Anemia is one of the most prevalent and pervasive nutritional (iron, folate, and vitamin A) deficiency disorders in the world, and it is a chronic condition that affects women of reproductive age [1,4–6] WHO) When a pregnant woman's Hgb concentration falls below 11 gm/dl, she becomes anemic [7]. It is estimated that 38.2% of pregnant women worldwide are affected by it, with South East Asia and Africa having the highest prevalence (48.7% and 46.3%, respectively), the Eastern Mediterranean region having a medium prevalence of 38.9%, and the Western Pacific, the Americas, and Europe having the lowest prevalence of 24.3%, 25.8%, and 24.9%, respectively [6]. It increased from 22% in Ethiopia in 2011 to 29% in 2016 [7].

Birth defects (BDs) are defined as structural, functional, behavioral, and metabolic flaws that appear at birth or are discovered later in life and develop during the organogenesis phase [8–10]. Over 90% of the estimated 8 million babies who are born each year with a serious birth defect live in LMICs. Birth defects affect every nation in the world. but LMICs bear a disproportionately heavy burden [11]. This demonstrates that regardless of age, race, income, or place of residence, all parents run the risk of having a child with a birth defect [12]. An estimated 303,000 of the 7.9 million children who are born each year with major congenital anomalies die within the first month of their life [10,13]. Congenital anomalies also cause 3.2 million children to be disabled [14]. Birth defects affect one in every 33 babies born in the United States, which is a significant factor in infant mortality [15]. In LMICs, more than 90% of congenital anomalies are found [16]. The prevalence of birth defects per 1000 live births varied significantly among South East Asian regions, from 54.1–64.3 [17]. Entebbe, Uganda, which is in Sub-Saharan Africa, had a prevalence of over 7% [13]. And 5.95% in Jimma, Ethiopia [18].

The WHO currently advises daily iron and folic acid supplementation as part of antenatal care to lower the risk of low birth weight, maternal anemia, and iron deficiency. The recommended supplement is said to contain 400 mg of folic acid, 30–60 mg of iron, with the higher dose preferred in areas where anemia in pregnant women is a serious public health issue (40%) [19]. Iron and folic acid supplementation can decrease the risk of maternal anemia and neural tube defects in offspring. Iron-folic acid supplementation of pregnant women increases Hgb levels in LMICs [20]. Deficiencies in iron and folic acid during pregnancy have a negative influence on the health of the mother, her pregnancy, as well as fetal development [21]. Maternal anemia during pregnancy has led to low weight gain, congestive heart failure, preterm labor, bleeding and lower resistance to infection, poor cognitive development, and reduced work capacity. Similarly, Folic Acid deficiency at conception and in early pregnancy has increased the risk of birth defects or neural tube defect, fetal malformations, and preterm delivery, puerperal sepsis, low birth weight, and preterm birth, birth asphyxia, neonatal mortality [22–32]. These risks are higher in LMIC [33]. In order to prevent and reduce these problems, adherence to iron and folic acid supplement is crucial and the most widely employed strategy globally at least by receiving 90 IFA supplements during Antenatal care [24,25,34].

The degree to which a patient complies with treatment recommendations made by a medical professional is known as adherence [35]. In order to achieve the goal of a 50% reduction in anemia in women of reproductive age by 2025 [22], the WHO defined the iron and folic acid supplementation (IFAS) indicator as "the proportion of women with a birth in the last 2 years

who received or bought for at least 6 months during their last pregnancy, in amounts that were in accordance with recommended protocols" [36]. The World Health Organization suggests that "all pregnant women should be counselled on the importance of adhering to a universal daily oral intake of one Iron/Folic Acid supplement (30 to 60 mg of elemental iron and 400 g (0.4 mg folic acid) supplement for six months (180 tablets in total) before conception and during early pregnancy [37] before delivery [25,38].

The updated data sources for the WHO IFAS coverage indicator were unable to provide all the details as they are currently defined. Worldwide, 70% of pregnant women did not take the recommended 180+ tablets, or 30 to 60 mg of iron and 400 g (0.4 mg) of folic acid, every day for six months. Only 8% of pregnant women in 22 Demographic and Health Survey countries did so [36,39,40]; In Pakistan, only about a quarter of women aged 15 to 49 reported taking IFA supplements for 90 or more days or not at all in the previous five years [41] and only 28% of pregnant women in Iran taking iron supplements as recommended time [42]. A study conducted in 22 Sub-Sahara African countries revealed that the overall prevalence of adherence to ≥ 90 days of iron supplementation during pregnancy was 28.7%, ranging from 1.4% to 73.0% in Burundi, Senegal, and Kenya it was a 32.7% [43–45]. In Ethiopia, 58% of women who gave birth to a live child in the previous five years did not take any iron-folic acid supplements during their most recent pregnancy in 2016, and this number fell to 40% in 2019. The percentage of pregnant women who took IFA supplements for more than 90 days rose from 5% in 2016 to 11% in 2019 [7,32] however, if they take the supplement at least four days a week, they are considered to have adhered to the IFAS [46].

In Ethiopia, roughly 74% of women had at least one ANC visit from a trained provider for their most recent live birth, and 4 in 10 women (43%) had four or more ANC visits [32]. The majority of the population is rural and spread out over vast areas without many roads, so they lack access to basic health services. In addition, there are weak health care delivery systems in place [47]. The care given to the mother before, during, and after pregnancy, childbirth, and the postpartum period has a significant impact on the health and survival of the newborn [48].

Numerous studies have demonstrated the relationship between various factors and pregnant women's non-adherence to IFAS. Pregnant women were less likely to take iron and folate supplements for a variety of reasons, including the side effect of the supplement forgetfulness [42,49–52], low awareness of the importance of IFAS use [53–55], big size tablets [56], too many tablets [26], inadequate awareness of anemia [54,57] and not receiving information about the necessity of IFA supplementation [57].

The National Nutrition Strategy targeted reducing the prevalence of Iron Deficiency and gave due attention to malnutrition in vulnerable groups of the society, particularly under-5 children, pregnant women, and nursing others. From 26.6% to 15% of women of childbearing age have anemia [58]. In order to prevent this and improve pregnant mothers' nutritional status, nutrition is incorporated into the health sector transformation plan as micronutrient interventions [59]. As soon as after the first month of gestation or at the time of the first ANC follow-up, all pregnant women should receive and consume 60 mg of elemental iron and 400 g of folic acid for 6 months during pregnancy and 3 months during postpartum, according to the national implementation guideline [57]. However, improvement in IFA supplementation and anemia control and prevention remains low and inadequate [60] because of poor maternal adherence to IFAS [34].

A mechanism to gauge adherence is required; it should be pertinent, doable, and reasonable. Adherence is a significant issue for IFA supplementation programs' effectiveness in Ethiopia. Only a small amount of research has been conducted and published under this heading in Eastern Ethiopia, particularly in the study area. Furthermore, the majority of earlier studies in Ethiopia only used quantitative research, making it difficult to evaluate cultural and

religious perspectives on the use of IFA supplements. To close this knowledge gap, it is essential to assess the adherence to iron and folate supplementation and associated factors among women attending antenatal care in public health facilities in Dire Dawa, Ethiopia.

## Methods and materials

### Study area and period

The study was conducted in Dire Dawa Administrative city which is located 515 km away from Addis Ababa. The town has 1 referral, 1 primary hospital, 3 private hospitals, 32 health posts, 2 clinics and 32 mid-level clinics, and 15 health centers. Based on the 2007 Census conducted by the Central Statistical Agency of Ethiopia (CSA), Dire Dawa has a population of 341,834, of whom 171,461 are men and 170,461 women. 233,224 or 68.23% of the population are urban inhabitants. The total estimated number of pregnant mothers who were attended in randomly selected nine health facilities (Dil Chora hospital, Goro HC, Lege Hare HC, Gende Kore HC, Melka Jebdu HC, Jelo Belina HC, Biyo Awale HC, Melka Kero HC, and Wahil HC) for a single month was 1011 [61]. A facility-based cross-sectional study supplemented by a phenomenological qualitative study design was employed from October 20–to November 19, 2020.

### Sampling size determination

The sample size for the quantitative study was determined by using the single population proportion formula for the first objective (prevalence) n = (Zα/2)2 p (1-p)/d2 by considering the following assumptions;

   *Where*:—**n** = minimum sample size required for the study

   **Z $_{\alpha/2}$** = standard normal distribution (Z = 1.96) with confidence interval of 95%

   **P** = prevalence of IFAS adherence in Dire Dawa, Ethiopia(p = 76%) [62].

   **d =** is a tolerable margin of error (d = 0.05)

   For the second objective (factors associated with IFAS, advised about IFAS) we have used double population proportion formula by considering CL:95%, Power:80%, Ratio:1:1, Exposed = 55.3%, Not exposed = 37.4% and became 264.The sample sizes for first objective is larger than that of the second objective. Hence, we took the larger sample size, which is 280. Accordingly, the final sample size came up by adding 10% non-response rate to the larger sample size. 280 x 10% = 28. 280 + 28 = 308. The final sample size was 308.

   The sample size for the qualitative study was determined based on the saturation of information collected from the study participants. Eight key informants (4 ANC focal persons and 4 health professionals) who had been working in the ANC clinic and four pregnant mothers who were not involved in the quantitative study were undergone In-Depth Interview (IDI).

### Sampling technique and procedure

Nine public health facilities (Dil chora hospital, Goro HC, Lege Hare HC, Gende Kore HC, Melka Jebdu HC, Jelo Belina HC, Biyo Awale HC, Melka Kero HC, and Wahil HC) were included using simple random sampling. The total estimated number of pregnant women attending antenatal clinic in the selected public health facilities for a single month is 1011 [61]. Sampling with population proportional to size was calculated for each health facility to give the total sample size by using the following formula.

$$\text{Proportionate allocation} = nj = n\ Nj/N$$

Simple and systematic random sampling techniques were used to select study participants.

A case review of previous pregnant mothers who attended antenatal clinics in the last month in selected health facilities was 1011. Hence, every three pregnant mothers ($K^{th}$) = N /n, 1011/ 308) were selected to be included in the study. The first pregnant mother was selected by lottery method, among the first 1 to 3 pregnant mothers who attended ANC follow up and the second pregnant woman was selected by lottery method from whom the first data was collected to select the $3^{rd}$ subjects until the sample size is fulfilled.

The qualitative study was applied as a supplement for the study to address issues which were important and could not be touched by the quantitative study. From the selected public health facilities, pregnant women were selected by using purposive sampling technique who were not participated in the quantitative study and focal persons of ANC clinic and technical person in ANC clinic were included using.

## Data quality control

The study questionnaire was developed after reviewing different literature and made available in English to be read, then translated back to the local languages of Amharic, Afan Oromo, and Somali, and finally translated back to English. Before data collection, a pretest study was done on 5% of the sample size in other health facilities, and an adjustment was made before using it for actual data collection.

Nine nurses for data collection and two BSc midwives for supervision were involved. Data collectors and supervisors received training for one day focused on interview technique, ethical issues, and rights of the participants, and they employed the interview guide during the discussion. The data collector interviewed the women in a private setting before visiting the ANC service on average for 30 minutes. In order to collect the qualitative data, the principal investigator conducted in-depth interviews using the IDI guide for pregnant women and health professionals separately. Investigator The principal investigator used a tape recorder to record the interview and filed notes on important points.

Appropriate personal protective equipment (PPE) were used which was adopted from WHO in the face of Covid-19 pandemic. Like sanitizer and a facemask was used by the data collectors and provided to the study participants before undergoing data collection. The data collectors used latex disposable gloves while taking information from the ANC card of the study participant in a clinical setting.

## Data instrument and processing

Data on socio-demographic information (age, marital status, religion, residence, maternal and paternal occupation, educational status, family size). Obstetrics and health-related characteristics (Gravida, parity, birthplace, first registration time for ANC, current visit trimester, ANC visit, current anemia status, etc.), health facility-related factors (time to take from home to facility, average stay time in the facility, etc.), and awareness about anemia and birth defects were collected.

Adherence can be categorized as having good adherence when pregnant women had taken at least 4 IFAS tablets per week in the previous month preceding the survey and those who took < 4 IFA tablets per week were categorized as having poor adherence [34,46,63–73].

The assessment of anemia awareness included 25 items. Each item received a score of "1" for the correct response and "0" for the incorrect response. Items were added up and converted to 100. If respondents scored above the median, they were classified as having good awareness of anemia; however, if they scored below the median, they were classified as having poor awareness of anemia [74].

Comprehensive awareness of birth defects was assessed by 12 questions. A right answer for each item was scored as "1" and a wrong answer was scored as "0." Items were totaled and given a score from 0 to 12. Respondents would be classified as having poor awareness of birth defects if they received a score of 0 to 4, moderate awareness if they received a score of 5–8, and good awareness if they received a score of 9 to 12 [69,75].

Nine items are used to evaluate an individual's overall understanding of iron-folate. Each item received a score of "1" for the correct response and "0" for the incorrect response. Items were added up and converted to 100%. If a respondent's score was above the median, they were classified as having good awareness of IFAS; if it was below the median, they were classified as having poor awareness of IFAS [74].

## Data analysis and interpretation

**For quantitative study.** Before data analysis, data was cleaned, coded, checked for normality and completeness, and entered into Epi data version 7.2.2.6 to minimize logical errors and design skipping patterns and then exported to SPSS version 25. The results are presented in tables and text using frequency and summary statistics such as mean, standard deviation, and percentage. Most of the variables were fitted to the bivariate analysis to obtain OR and the CI for an association of variables. All variables with $p \leq 0.25$ in the bivariate analysis were further entered into a multivariate logistic regression model in order to control all possible confounders, and standard enter techniques were fitted. The variables having a p-value $< 0.05$ in the multivariate analysis were taken as statistically significant predictors of adherence. The Hosmer and Lemeshow goodness-of-fit tests were used to assess whether the necessary assumptions for the application of multiple logistic regression were fulfilled and a p-value $> 0.05$ was considered a good fit.

**For the qualitative study.** To understand the perceptions of behavioral and cultural issues in terms of motivating factors and impediments to participants' use of antenatal IFA supplementation, the principal investigator had undergone in-depth interviews with guiding questions. Data manually analyzed following meticulous recording of the interview. The field notes were expanded in the language used by the study participants, and the audio recordings and notes were translated into English by the investigator and a native speaker of the respondents' language. The transcriptions of the audio recordings and discussions were done verbatim. Following the distribution of codes, themes and issues were developed. The text related to each code/theme was then discussed and condensed into a document that presented the results using quotes. The codes were created from the texts, and they were divided into the following categories: the availability of tablets, knowledge of IFA supplementation, attitudes/opinions toward IFAs, advice from healthcare professionals, and how women are perceived. The themes that emerged from the codes would serve as supplements for the benefits of or obstacles to taking iron and folate supplements. Each transcript was thoroughly examined, and the quantitative outcome was triangulated.

**Ethical consideration.** A formal letter of cooperation was written to the Dire-Dawa city administration health bureau and to each individual health facility after the Dire Dawa University Ethics Review Committee granted ethical clearance. While conducting in-person interviews, data collectors and study participants safeguarded themselves by being concerned about their exposure risk. Women who had signed the informed consent form and agreed to participate were interviewed. All of the interviewees received information about the use of antenatal IFA supplementation, its effects on maternal health, and the growth and development of the newborn at the conclusion of each interview session from the data collectors. Personal identifiers were excluded from the study to ensure confidentiality.

# Result

## Socio-demographic characteristics

A total of 308 pregnant women took part in the study, yielding a response rate of 100%. The study's participants were 27.08 years old on average (±5.59 SD). Three fifths (64.3%) of pregnant women and their partners were housewives, and 124 (40.3%) of their partners were private employees. All of the study participants (100%) were married; 65.3% were Muslims; 92 (29.9%) of pregnant women and 141 (45.8%) of their partners had at least a high school diploma. Urban residents made up more than two-thirds (71.1%) of the respondents, and more than half (54.2%) had four or more family members [**Table 1**].

**Table 1. Socio-demographic characteristics of pregnant women attending antenatal care in public health facilities Dire Dawa, Eastern Ethiopia. n = 308.**

| Variables | Frequency (%) |
| --- | --- |
| **Age** | |
| 15–24 | 98 (31.8) |
| 25–34 | 168 (54.5) |
| ≥ 35 | 42 (13.6) |
| **Marital Status** | |
| Married | 308 (100) |
| **Religion** | |
| Orthodox | 85 (27.6) |
| Muslim | 201 (65.3) |
| Other** | 22 (7.1) |
| **Mother level of education** | |
| No formal education | 141 (45.8) |
| Primary education | 75 (24.4) |
| Secondary and above | 92 (29.9) |
| **Occupational status of mother** | |
| Housewife | 198 (64.3) |
| Government employee | 26 (8.1) |
| Self-employee | 62 (20.1) |
| Other*** | 23 (7.5) |
| **Husband level of education** | |
| No formal education | 117 (38) |
| Primary education | 50 (16.2) |
| Secondary and above | 141 (45.8) |
| **Husband occupational status** | |
| Government employee | 59 (19.2) |
| Private employee | 124 (40.3) |
| Daily laborer | 62 (20.1) |
| Farmer | 49 (15.9) |
| Other**** | 14 (4.5) |
| **Family size** | |
| 1–3 | 141 (45.8) |
| ≥ 4 | 167 (54.2) |
| **Place of residence** | |
| Urban | 219 (71.1) |
| Rural | 89 (28.9) |

Other*: Divorced, Widowed, Single Other**: Protestant, Catholic, other***: Daily laborers, Farmer, student, other****: No job, Student.

## Obstetrics and health related characteristics

The pregnant women at the time of the current visit had a mean gestational age of 30.63 (SD 5.91) weeks, and 238 (77.3%) of them were in the third trimester. 128 (41.6%) of the pregnant women who were registered for the first visit had gestational ages of 16 weeks or less. The mean gestational age of the pregnant women at their first visit was 15.96 (±SD 6.68) weeks. Mothers made up more than four-fifths (81.2%) of the population, and 82 (26.6%) of them were primiparous. According to this study, 97 (31.5%) of the pregnant women had anemia. Four-fifths of respondents (79.9%) had fewer than four ANC visits [Table 2].

**Table 2. Obstetrics and health related characteristics of pregnant women attending antenatal care in public health facilities Dire Dawa, Eastern Ethiopia. n = 308.**

| Variables | Adherence | | Frequency (%) |
|---|---|---|---|
| | Yes (%) | No (%) | |
| **Gravidity** | | | |
| Primigravida | 34 (58.6) | 24 (41.4) | 58 (18.8) |
| Multigravida | 140 (56) | 110 (44) | 250 (81.2) |
| **Parity** | | | |
| Nullparous | 35 (57.4) | 26 (42.6) | 61 (19.8) |
| Primiparous | 56 (68.3) | 26 (31.7) | 82 (26.6) |
| Multiparous | 83 (50.3) | 82 (49.7) | 165 (53.6) |
| **Birth spacing in years** | | | |
| 1–2 | 63 (43.4) | 82 (56.6) | 145 (58.7) |
| ≥ 3 | 76 (74.5) | 26 (25.5) | 102 (41.3) |
| **ANC visit** | | | |
| < 4 visits | 123 (50) | 123 (50) | 246 (79.9) |
| ≥ 4 visits | 51 (82.3) | 11 (17.7) | 62 (20.1) |
| **First registration time for ANC** | | | |
| Early registration | 89 (69.5) | 39 (30.5) | 128 (41.6) |
| Late registration | 85 (47.2) | 95 (52.8) | 180 (58.4) |
| **Current visit trimester** | | | |
| First and second trimesters | 40 (57.1) | 30 (42.9) | 70 (22.7) |
| Third trimester | 134 (56.3) | 104 (43.7) | 238 (77.3) |
| **Medical illness other than anemia** | | | |
| No | 152 (56.9) | 115 (43.1) | 267 (86.7) |
| Yes | 22 (53.7) | 19 (46.3) | 41 (13.3) |
| **Anemia during previous pregnancy** | | | |
| No | 129 (58.4) | 92 (41.6) | 221 (71.8) |
| Yes | 45 (51.7) | 42 (48.3) | 87 (28.2) |
| **Current anemia status** | | | |
| Normal | 123 (58.3) | 88 (41.7) | 211 (68.5) |
| Anemic | 51 (52.6) | 46 (47.4) | 97 (31.5) |
| **Gestational age during Hgb test** | | | |
| First trimester | 85 (74.6) | 29 (25.4) | 114 (37) |
| Second and third trimesters | 89 (45.9) | 105 (54.1) | 194 (63) |

ANC, Antenatal care; Hgb, Hemoglobin.

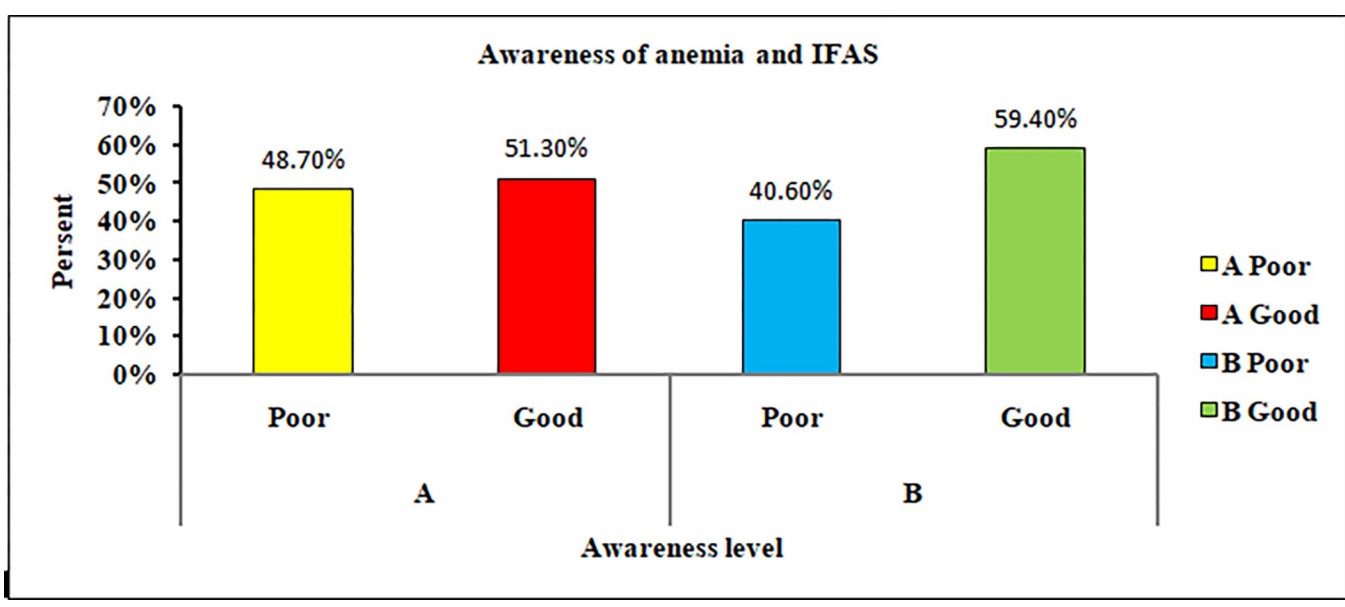

**Fig 1. Awareness of anemia and Iron-folate supplementation among pregnant women attending antenatal care in public health facilities at Dire Dawa, Eastern Ethiopia.** (A) Explains pregnant women level of awareness of anemia; (B) Explains pregnant women level of awareness of iron folic acid supplements.

## Awareness status of respondents on anemia, birth defect and IFAS

Of the respondents, 183 (59.4%) had good awareness of iron and folic acid supplementation, and 158 (51.3%) had good awareness of anemia, Half of respondents (50.6%) had a low level of awareness about birth defects "**Figs 1 and 2**".

## Health facility related characteristics

194 respondents (63%) said they waited an average of less than or equal to 30 minutes to receive ANC services in the health facilities, and 203 respondents (65.9%) said it took them less than 30 minutes to get to the nearest medical facility. 266 (86.4%) of respondents received

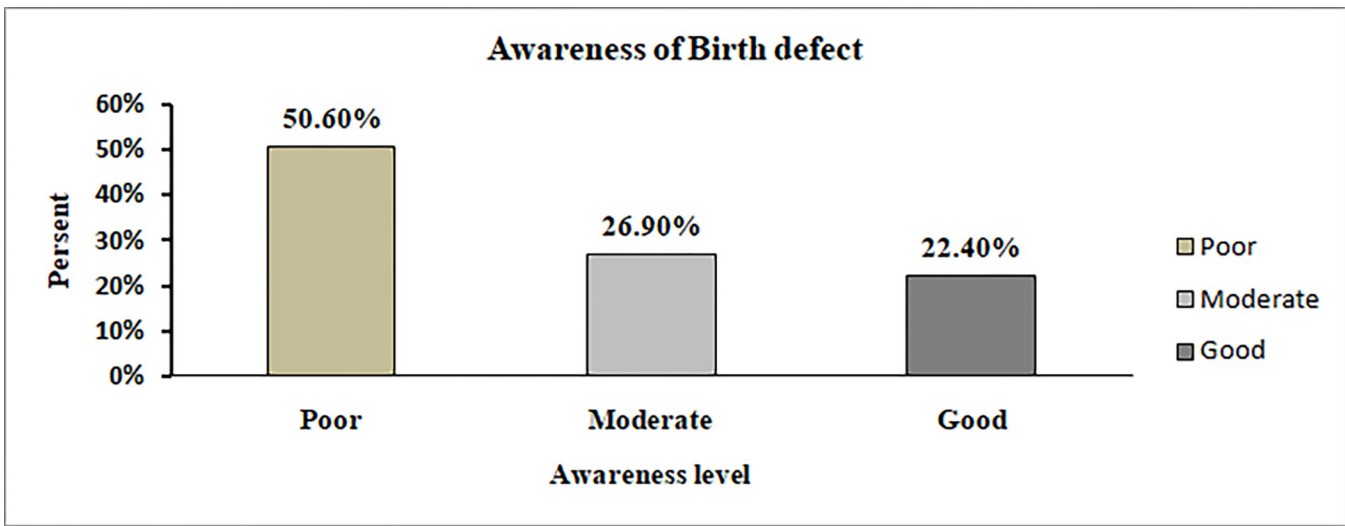

**Fig 2. Awareness level of birth defect among pregnant women attending antenatal care in public health facilities at Dire Dawa, Eastern Ethiopia.**

**Table 3. Health facility related characteristics of pregnant women attending antenatal care in public health facilities at Dire Dawa, Eastern Ethiopia. n = 308.**

| Variables | Adherence | | Frequency (%) |
|---|---|---|---|
| | No (%) | Yes (%) | |
| **Time taken from home to facility** | | | |
| ≤ 30 minutes | 96 (47.3) | 107 (52.7) | 203 (65.9) |
| > 30 minutes | 38 (36.2) | 67 (63.8) | 105 (34.1) |
| **Average stay time in the facility** | | | |
| ≤ 30 minutes | 82 (42.3) | 112 (57.7) | 194 (63) |
| > 30 minutes | 52 (45.6) | 62 (54.4) | 114 (37) |
| **Got counseling on IFA** | | | |
| No | 22 (55) | 18 (45) | 40 (13) |
| Yes | 112 (41.8) | 156 (58.2) | 268 (87) |
| **Faced shortage of supplement** | | | |
| No | 113 (42.6) | 152 (57.4) | 265 (86) |
| Yes | 21 (48.8) | 22 (51.2) | 43 (14) |
| **Got IFA without payment** | | | |
| No (from private pharmacy) | 20 (47.6) | 22 (52.4) | 42 (13.6) |
| Yes | 114 (42.9) | 152 (57.1) | 266 (86.4) |

IFA; Iron and folate supplements.

free IFA tablets, 265 (86%) experienced no shortage of supplements during their ANC visit, and 268 (87%) of respondents received counseling on IFA tablets [**Table 3**].

## Self-reported adherence status to IFA Supplementation

Pregnant women who attended antenatal clinics in public health facilities overall self-reported adherence status (took IFA supplement for > = 4 days/week for the previous one month prior to the survey) was 174 (56.5%) "**Fig 3**".

## Reasons for adherence and barriers of adherence to IFAS

**Reasons for adherence to IFAS.** Based on the quantitative findings, among all respondents who visited antenatal clinic, 94 (54%),had adhered to iron and folic acid supplements, because of the counselling they received a counselling from care provider, followed by 78 (44.8%), who said they were afraid of the consequences if they stopped taking the supplements. In the opinion of 58 (33.3%) respondents, these supplements would result in an increase in blood volume, and 45 (25.8%) said this was because of the support of family. 94 respondents claimed they continued taking the IFA supplement because they sought out clinical advice regarding it. A qualitative study backed up this conclusion.

*During her fourth visited at Dilchora Hospital for ANC follow-up, a 25-year-old pregnant woman said, "My sister was a health professional." She advised me to visit a medical facility and consult a medical expert about the IFA supplement. I took her advice and immediately began taking the supplement.*

*Another pregnant 32-year-old who visited LegeHare HC for an ANC checkup stated, "The doctor told me to consume the supplement three times per day, otherwise I would face anemia." I regularly took the supplement because of this.*

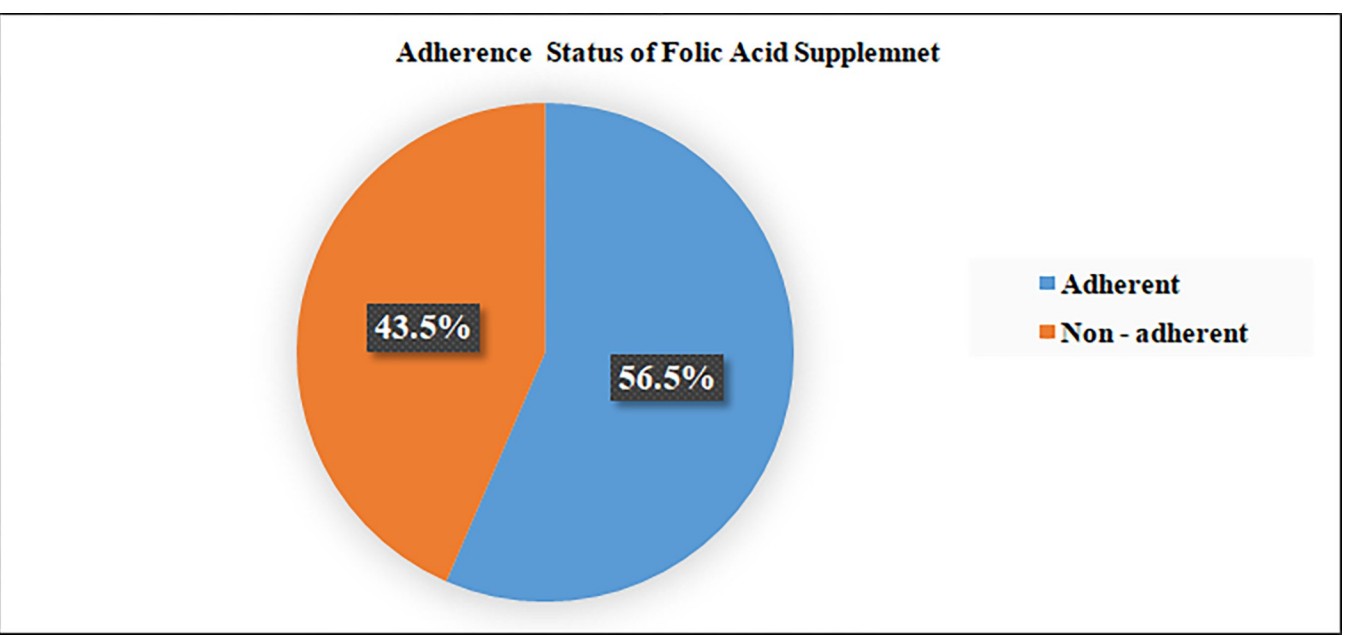

**Fig 3. Level of adherence status of iron folic acid supplement among pregnant women attending public health facilities at Dire Dawa, Eastern Ethiopia.**

*A pregnant 31-year-old woman who visited Goro Health Center for ANC follow-up said, "I took the supplement regularly because I was afraid that I would experience anemia in the future if I did not consume it."*

Fifty-eight of the respondents adhered to the IFA supplement because they thought the supplement would increase their blood. This finding was supported by a qualitative study.

*When a 25-year-old pregnant woman visited Gende Kore HC for ANC follow-up, she said, "The doctor told me that I had anemia and I had to take the supplement strictly." She responded, "I took the tablet for three consecutive months without stopping because I thought the supplement would increase my blood volume," when I asked why.*

**Barriers of adherence to IFAS.** The main obstacle to taking IFA supplements, according to more than half of pregnant women (58.2%), was experiencing side effects, followed by forgetfulness 59 (44%). In health facilities, 43 (32%) reported a shortage of supplements; 35 (26.1%) thought that taking too many pills would be harmful to the mother and/or her unborn child; and 28 (21%) were unaware of the significance of the supplements. The side effects that 78 pregnant women claimed they experienced prevented them from taking the IFA supplement.

A qualitative study backed up this conclusion.

*The 22-year-old expectant mother who attended the ANC appointment at Jelo Belina HC stated, "I had experienced nausea and a burning sensation around the epigastric site while consuming the supplement. For this reason, I stopped taking the supplement.*

*The health facility cleaner reported to a 29-year-old midwifery and MCH focal person that she got so many IFA supplements thrown in garbage and slum areas, He also added, "Most*

*pregnant women did not want to encounter any side effects of the supplement, and they would not consume the supplement unless they had anemia."*

49 mothers admitted that they had forgotten to take the supplement on a regular basis. A qualitative study backed up this conclusion.

*I would take the IFA supplement as soon as I remembered to take it, as an 18-year-old primigravida who attended the ANC follow-up clinic at Gende Kore health center said to me, "I forgot to take the supplement every day.*

Forty-three pregnant women suggested that one of the reasons they did not take their IFA supplements regularly was because there were shortage of IFA supplements in the health facilities. This finding was supported by findings from the qualitative study.

There has been a shortage of iron and folate supplements in the hospital for almost six months, according to a 28-year-old nurse who serves as the MCH focal point at Dil Chora Hospital. She also stated that "*we sent all pregnant women to buy the supplement at their own expense from private pharmacies found outside the hospital."*

A 27-year old midwife working as MCH focal person at Jelo Belina Health Center told me,"... *"The health center encountered a shortage of IFAS because it had not received the supplement from EPSA directly, but rather it had received it indirectly from other health centers found in the urban area, Addis Ketema Health Center,"*

Concerning reported side effects, more than three fifths of participants report having developed gastric upset 62 (79.5%), followed by heartburn 48 (61.5%), and nausea and vomiting 35 (44.9%) "**Fig 4**'.

**Factors associated with adherence to IFA supplementation.** The following covariates were candidates for the multivariable model: mother's education level, husband's education level, family size, residence, counseling sought, ANC visit, first registration week, knowledge of anemia, knowledge of IFA, and knowledge of birth defect. Mother's education, place of residence, ANC visit, first registration week, and awareness of birth defects were linked to adherence to iron and folic acid supplementation in the multivariable model. Primary-educated mothers were 62% less likely to follow IFAS than those with secondary and higher education (AOR = 0.38, 95% CI: 0.17–0.83). In comparison to mothers who lived in rural areas, mothers

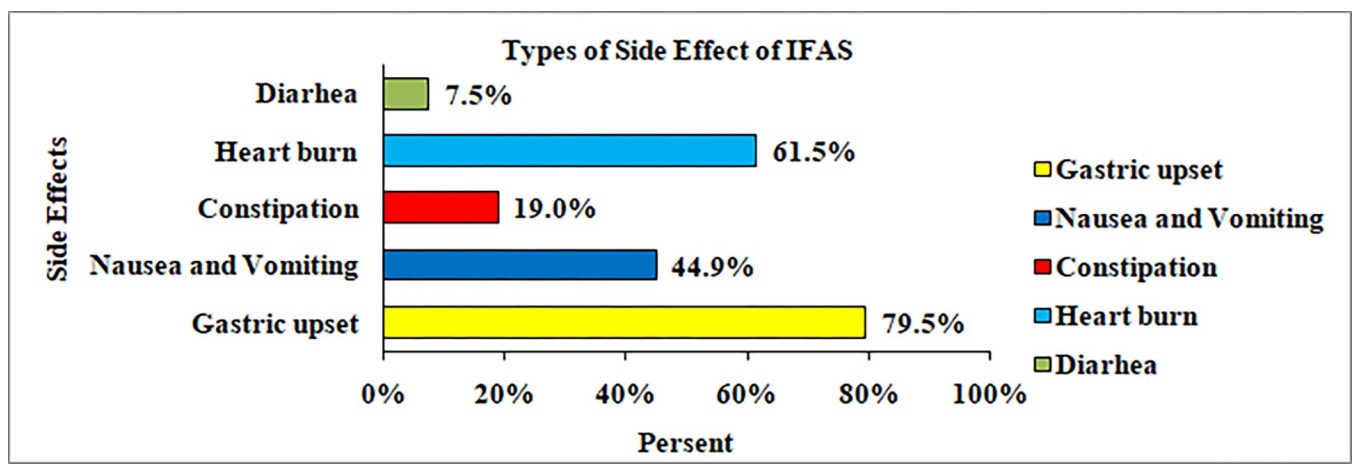

**Fig 4. Reported side effects of Iron and folic acid supplementation among pregnant women attending antenatal care in public health facilities at Dire Dawa, Eastern Ethiopia.**

**Table 4. Factors associated with adherence to iron and folate supplementation among pregnant women attending antenatal care in public health facilities at Dire Dawa, Eastern Ethiopia. n = 308.**

| Variables | Adherence | | COR(95%CI) | AOR(95%CI) |
|---|---|---|---|---|
| | No (%) | Yes (%) | | |
| **Mother's level of education** | | | | |
| No formal education | 66 (46.8) | 75 (43.1) | **0.40 (0.23–0.71)**\*\* | 1.50 (0.62–3.65) |
| Primary education | 44 (58.7) | 31 (41.3) | **0.25 (0.13–0.48)**\*\*\* | **0.38 (0.17–0.83)**\* |
| Secondary and above | 24 (26.1) | 68 (73.9) | 1 | 1 |
| **Residence** | | | | |
| Urban | 78 (35.6) | 141 (64.4) | 1.55(0.93–2.58) | **4.61 (2.19–9.65)**\*\*\* |
| Rural | 42 (47.2) | 47 (52.2) | 1 | 1 |
| **ANC visit** | | | | |
| ≥ 4 visits | 11 (17.7) | 51 (82.3) | **4.64 (2.31–9.32)**\*\*\* | **3.05 (1.23–7.58)**\* |
| < 4 visits | 123 (50) | 123 (50) | 1 | 1 |
| **First registration time for ANC** | | | | |
| Early registration | 39 (30.5) | 89 (69.5) | **2.55 (1.58–4.11)**\*\*\* | **4.01 (1.66–9.69)**\*\* |
| Late registration | 95 (52.8) | 85 (47.2) | 1 | 1 |
| **Knowledge birth defect** | | | | |
| High/ | 13 (18.8) | 56 (81.2) | **4.42 (2.24–8.73)**\*\*\* | **5.66 (1.85–17.27)**\*\*\* |
| Moderate | 42 (50.6) | 41 (49.4) | 1.02 (0.59–1.71) | 0.72 (0.31–1.69) |
| Low/ | 79 (50.6) | 77 (49.4) | 1 | 1 |

\*P≤0.05

\*\*P≤0.01

\*\*\*P≤0.001.

who lived in urban areas were 4.61 times (AOR = 4.61, 95% CI: 2.19–9.65) more likely to adhere to IFAS. Mothers who attended four or more ANC visits were 3.05 times (AOR = 3.05, 95%CI: 1.22–7.58) more likely to adhere to IFAS than their counterparts were. Mothers who registered early were 4.01 times (AOR = 4.01, 95%CI: 1.66–9.69), and mothers who were well-informed about birth defects were 5.66 times (AOR = 5.66, 95%CI: 1.85–17.27) more likely to do so [**Table 4**].

## Discussion

This study found that the self-reported adherence rate of the pregnant women was 56.5% (95% CI: 51%–62.2). This finding is consistent with the study conducted in Nepal (58%) [76], Senegal (51%) [45], North West Ethiopia (55.3%) [64], Oromia, Ethiopia (59.8%) [52]; Southern Ethiopia (51.4%) [49], and Amhara, Ethiopia (52.9%) [71].

However, the finding was higher than other studies conducted in Kiambu country, Kenya (32.7%) [49], Mekelle, Tigray, Ethiopia (10.5%) [77], Western Amhara, Ethiopia (20.4%) [26], South East Ethiopia (18%) [54] and Afar, Ethiopia (22.9%) [69]. The reason behind the difference may be that most of the pregnant women involved in this study were from urban areas, and the pregnant women might get information from different sources like the media and health centers. The other possible explanation could be that the adherence might have been overestimated as it is calculated based on the self-report of the mothers.

On the other hand, the adherence status found in this study is lower than in other studies done in Iran (71.6%) [42], Eritrea refugee camp, Tigray, Ethiopia (64.7%) [57], Dire Dawa, Ethiopia 76% in 2018 and 71.8% in 2019 [62,78], and Southern Ethiopia (70.6%) [72].

This might be due to most pregnant mothers did not attend ANC follow up properly due to fear of contracting the COVID-19 from health facilities. The other possible explanation might be the fact that there was scarcity of the IFAS for a longer period in some health institutions, especially in Dil Chora Hospital, where the supplement was unavailable for more than 6 months. Low percentage of attending four or more ANC follow up no strict follow up by the regional health bureau, as there were no indicators to assess adherence to iron foliate supplementation in the Ethiopian health service reporting system and the present study included both rural and urban populations, which could be the other reason for the difference.

Mothers who had primary education were 62% less likely to adhere to IFAS than those who had secondary and above. This finding goes in line with other studies done in Senegal [45] and Southern Ethiopia [49,52].

This might be educated women have better access to information about iron deficiency anemia, therapy, pregnancy-related risk factors, and their consequences through reading books, the internet, seeking advice from healthcare professionals, and understanding the benefits of the supplement.

This study revealed that mothers who lived in urban areas were 4.61 times more likely to adhere to IFAS than mothers who lived in rural areas. The result was consistent with other studies conducted in Malawi [79], Afar, Ethiopia [68], and North West Ethiopia [64].

This difference can be better elaborated as urban dwellers are obviously living in better infrastructures like good access to transportation and getting to nearby health facilities than rural dwellers. This helps urban dwellers get health advice from health care professionals regularly.

The number of ANC visits was another predictor that had a significant impact on adherence. Mothers who had four or more ANC visits were three times more likely to adhere to IFAS than mothers who had less than four ANC visits. Other studies conducted in Indonesia [80], 22 Sub-Saharan African countries [43], Mulago National Referral Hospital, Uganda [51], Southern Senegal [45], Amhara, Ethiopia [66,71], and Eritrean refugee camp, Tigray, Ethiopia [57] supported this finding.

This might be better exemplified as when pregnant women come to ANC follow up, they might get iron and folate supplementation with counseling from health care professionals. They might hear about the benefits of adherence to the table and the consequences of discontinuation of the supplement. For these reasons, they might have to strictly adhere to IFAS.

Gestational age at the time of first registration to ANC follow-up was another predictor that showed significant association with adherence to IFAS. Pregnant women who had early registration time were four times more likely to adhere than those who came late for registration. The finding is consistent with other studies conducted in the Amhara, Afar, SNNP, and Oromia regions of Ethiopia [46,52,64,66,68].

The possible explanation for the consistency might be pregnant mothers who came early for ANC follow-up had a greater concern for positive pregnancy outcomes. In addition, she would have a greater tendency to know their anemia status early and to attend more ANC visits throughout her pregnancy. This in turn might help her to get repeated counseling on the benefits of IFAS, take more IFA supplements, and adhere to IFA supplements.

Adherence has been better observed among pregnant women who had a good level of awareness about birth defects. Pregnant women who had good awareness about birth defects were 5.66 times more likely to adhere to IFAS as compared to those who had low knowledge about birth defects. This result goes in line with a study conducted in Indonesia which revealed that pregnant mothers who had good awareness of pregnancy-related risks were associated with two times increased adherence to IFA supplementation than those who had poor awareness of pregnancy-related risks [80].

The reason behind this might be as if a pregnant woman has awareness about pregnancy-related risk factors like birth defects due to folic acid deficiency, it will lead them to minimize the risks by taking IFA supplements. By taking the supplement, she might aware that she can protect herself and her baby from bad pregnancy outcomes. Moreover, there were no previous other studies that suggested pregnant women's knowledge about birth defects was a predictor of adherence to IFAS.

## Limitation of the study

The gold standard methods for assessing adherence, such as electronic and pill counting methods were not used in this study because they were either prohibitively expensive or unavailable. The study used a self-reported method to assess the adherence status of pregnant women. In addition, the study might expose the mother to recall bias since the study assessed the mother's adherence status in the previous one month.

## Conclusion

In comparison to earlier studies conducted in the study area, the study's results show that pregnant women adhere to iron and folate supplements at relatively low rates. The mother's educational level, the location, the number of ANC visits, the gestational age for the first registration of ANC, and the pregnant women's awareness of birth defects were predictors for adherence to iron and folate supplements. The main justifications for adherence to IFAS were cited as receiving clinical counseling about the supplement, being afraid of the repercussions if they did not take it, and believing that the supplement would increase their blood volume. Contrarily, pregnant women reported adverse effects of the supplement, forgetfulness, and difficulty finding the supplement in medical facilities.

## Acknowledgments

We would like to thank Dire-Dawa University, College of Medicine and Health Sciences, and IRB for their technical support and for giving us clearance to conduct this study. We are also grateful to our colleagues, who in one way or another helped us to undertake this research. A great deal of gratitude goes to data collators and study participants, especially.

## Author Contributions

**Conceptualization:** Arayasillase Assegid Tefera, Neil Abdurashid Ibrahim, Abdurezaq Adem Umer.

**Data curation:** Arayasillase Assegid Tefera, Neil Abdurashid Ibrahim.

**Formal analysis:** Arayasillase Assegid Tefera, Neil Abdurashid Ibrahim, Abdurezaq Adem Umer.

**Funding acquisition:** Arayasillase Assegid Tefera.

**Investigation:** Neil Abdurashid Ibrahim, Abdurezaq Adem Umer.

**Methodology:** Arayasillase Assegid Tefera, Neil Abdurashid Ibrahim, Abdurezaq Adem Umer.

**Validation:** Arayasillase Assegid Tefera, Neil Abdurashid Ibrahim, Abdurezaq Adem Umer.

**Visualization:** Abdurezaq Adem Umer.

**Writing – review & editing:** Neil Abdurashid Ibrahim.

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
