## [Decision Letter · Decision Letter 0]

22 Sep 2022

PGPH-D-22-01055

ADHERENCE TO IRON AND FOLATE SUPPLEMENTATION AND ASSOCIATED FACTORS AMONG WOMEN ATTENDING ANTENATAL CARE IN PUBLIC HEALTH FACILITIES AT COVID-19 PANDEMIC IN ETHIOPIA

Dear Dr. IBRAHIM,

Thank you for submitting your manuscript to PLOS Global Public Health. After careful consideration, we feel that it has merit but does not fully meet PLOS Global Public Health’s publication criteria as it currently stands. Therefore, we invite you to submit a revised version of the manuscript that addresses the points raised during the review process.

The manuscript has been evaluated by two reviewers, and their comments are available below.

The reviewers have raised a number of concerns that need attention, and they request additional information on methodological aspects of the study and analyses.

Could you please revise the manuscript to carefully address the concerns raised?

We look forward to receiving your revised manuscript.

Kind regards,

Vanessa Carels

Staff Editor

Journal Requirements:

2. Please amend your Data Availability Statement and indicate where directly the data may be found.

Additional Editor Comments (if provided):

Reviewers' comments:

Reviewer's Responses to Questions

**Comments to the Author**

1. Does this manuscript meet PLOS Global Public Health’s publication criteria? Is the manuscript technically sound, and do the data support the conclusions? The manuscript must describe methodologically and ethically rigorous research with conclusions that are appropriately drawn based on the data presented.

Reviewer #1: Yes

Reviewer #2: Partly

2. Has the statistical analysis been performed appropriately and rigorously?

Reviewer #1: Yes

Reviewer #2: Yes

3. Have the authors made all data underlying the findings in their manuscript fully available (please refer to the Data Availability Statement at the start of the manuscript PDF file)?

Reviewer #1: Yes

Reviewer #2: Yes

4. Is the manuscript presented in an intelligible fashion and written in standard English?

Reviewer #1: Yes

Reviewer #2: No

5. Review Comments to the Author

Reviewer #1: This is a very interesting mixed-methods study that aimed to assess adherence to iron and folate supplementation and its associated factors among women attending antenatal care in public health facilities in a city in Ethiopia. Although the importance of the topic, the present study presents several gaps.

1. First of all, the Background section is very disorganized. Therefore, for readability, it would be best to approach the content in the following logical sequence: (i) anemia and its consequences for the mother and baby; (ii) the importance of iron and folic acid supplementation; (iii) treatment adherence definition; (iv) proportions of adherence/non-adherence to IFAS worldwide and in the region; (v) exploration of the associated factors; (vi) study rationale; (vii) aims.

2. Background section – The phrase “More than 50% of pregnant women were anemic in Pakistan in 2011(38).” can be removed since the authors address other more up-to-date references.

3. Also, the correct aim of the study is to assess adherence to iron and folate supplementation and its associated factors among women attending antenatal care in public health facilities in Dire Dawa city administration, Eastern Ethiopia.

4. Methods section – The authors described the study area very well. However, they did not mention the estimated population size of the city.

5. Methods section – There was a lack of information about the variables used in the analysis, such as how they were classified; how they were collected (all by self-report?)...

6. Methods section – In the paragraph: “Appropriate personal protective equipment (PPE) like sanitizer and a facemask was used by the data collectors and provided to the study participants before undergoing data collection. The data collectors used latex disposable gloves while taking information from the ANC card of the study participant in a clinical setting.” it is important to state that those measures were adopted in the face of the COVID-19 pandemic.

7. Results section – “the educational status of pregnant women was 92 (29.9%)”: this information is incomplete.

8. Results section – “and 124 (40.3%) of their partner were government employees”: this information differs from the table.

9. Discussion section – Several statements are unreferenced: “This might be due to the fact that educated women have better access to information about iron deficiency anemia, therapy, pregnancy-related risk factors, and their consequences through reading books, the internet, seeking advice from healthcare professionals, and understanding the benefits of the supplement.”; “This difference can be better elaborated as urban dwellers are obviously living in better infrastructures like good access to transportation and getting to nearby health facilities than rural dwellers. This helps urban dwellers get health advice from health care professionals regularly.”; “The reason behind this fact is that if a pregnant woman has knowledge about pregnancy related risk factors like birth defects due to folic acid deficiency, it will lead them to minimize the risks by taking IFA supplements. By taking the supplement, she might aware that she can protect herself and her baby from bad pregnancy outcomes.”

10. There are some typos in the manuscript.

Reviewer #2: The topic described in this study is important and interesting. To improve this manuscript, please read the following suggestions:

Abstract:

1 Background: it is better to provide the actual dose than use the ”tab”.

2 Methods: the 1st sentence needs to be improved for readability; How was adherence defined?

3 Results: please indicate the high/low maternal education, low/high number of ANC visits, high/low gestational age, with/without knowledge about birth defects were associated with ...; please also indicate what categories were included in the place of residence variable.

Background

Please rearrange the background to increase readability and logic flow.

1 The sentence “However, most pregnant women in the world did not consume the full amount (180 tablets/ 6 months)” needs a reference.

2 Why we need “another” in the “Hence, there is a need for another mechanism to measure adherence, which should be relevant, attainable, and reasonable.” is not clear

3 Please double check the references for “Women are said to have adhered to IFAS if they take the supplement at least 4 days a week. (3–15).”, not all of them can support the statement.

4 Please provide a reference for the sentence “In Ethiopia, about 74% of women received ANC from a skilled provider at least once for their last birth and four in10 women (43%) had four or more ANC visits for their most recent live birth.”.

5 Please provide a reference for “In Ethiopia, about 74% of women received ANC from a skilled provider at least once for their last birth, and four in 10 women (43%) had four or more ANC visits for their most recent live birth.”

Methods:

1 Please provide more information about the choice of assumption for the calculation of sample size, for example, why did you assume the nonresponse rate to be 10%

2 Please revise the sample size calculation and sampling method to improve readability.

Results:

1 The overall structure of the result section needs to be improved. All the information in the result section should be connected and tell a clear story.

2 “This finding was supported by a qualitative study” is not clear. Please improve its readability.

Whole manuscript:

1. The grammar of the whole draft needs to be revised to improve readability. Most information communicated in this manuscript is not clear enough for the reviewer to provide suggestions with confidence.

6. PLOS authors have the option to publish the peer review history of their article (what does this mean?). If published, this will include your full peer review and any attached files.

**Do you want your identity to be public for this peer review?** For information about this choice, including consent withdrawal, please see our Privacy Policy.

Reviewer #1: No

Reviewer #2: No

---

## [Decision Letter · Decision Letter 1]

8 Dec 2022

ADHERENCE TO IRON AND FOLATE SUPPLEMENTATION AND ASSOCIATED FACTORS AMONG WOMEN ATTENDING ANTENATAL CARE IN PUBLIC HEALTH FACILITIES AT COVID-19 PANDEMIC IN ETHIOPIA

PGPH-D-22-01055R1

Dear Mr. IBRAHIM,

We are pleased to inform you that your manuscript 'ADHERENCE TO IRON AND FOLATE SUPPLEMENTATION AND ASSOCIATED FACTORS AMONG WOMEN ATTENDING ANTENATAL CARE IN PUBLIC HEALTH FACILITIES AT COVID-19 PANDEMIC IN ETHIOPIA' has been provisionally accepted for publication in PLOS Global Public Health.

Best regards,

Julia Robinson

Executive Editor

Reviewer Comments (if any, and for reference):

Reviewer's Responses to Questions

**Comments to the Author**

1. If the authors have adequately addressed your comments raised in a previous round of review and you feel that this manuscript is now acceptable for publication, you may indicate that here to bypass the “Comments to the Author” section, enter your conflict of interest statement in the “Confidential to Editor” section, and submit your "Accept" recommendation.

Reviewer #1: All comments have been addressed

2. Does this manuscript meet PLOS Global Public Health’s publication criteria? Is the manuscript technically sound, and do the data support the conclusions? The manuscript must describe methodologically and ethically rigorous research with conclusions that are appropriately drawn based on the data presented.

Reviewer #1: Yes

3. Has the statistical analysis been performed appropriately and rigorously?

Reviewer #1: Yes

4. Have the authors made all data underlying the findings in their manuscript fully available (please refer to the Data Availability Statement at the start of the manuscript PDF file)?

Reviewer #1: Yes

5. Is the manuscript presented in an intelligible fashion and written in standard English?

Reviewer #1: Yes

6. Review Comments to the Author

Reviewer #1: Congratulations to the authors for the excellent study.

A single note is that the description of the abbreviation for ANC is missing from the "Abbreviations" section.

7. PLOS authors have the option to publish the peer review history of their article (what does this mean?). If published, this will include your full peer review and any attached files.

**Do you want your identity to be public for this peer review?** For information about this choice, including consent withdrawal, please see our Privacy Policy.

Reviewer #1: No
